# The magnitude of non-adherence and contributing factors among adult outpatient with Diabetes Mellitus in Dilla University Referral Hospital, Gedio, Ethiopia

Bisrat Desalegn Boshe[1], Getachew Nenko Yimar[2], Aberash Eifa Dadhi[3], Worku Ketema Bededa[4]*

**1** Department of Internal Medicine, St Paul Millennium College, Addis Ababa, Ethiopia, **2** School of Public Health, College of Medicine and Health Science, Dilla University, Dilla, Ethiopia, **3** Department of Midwifery, College of Medicine and Health Science, Hawassa University, Hawassa, Ethiopia, **4** Department of Pediatrics and Child Health, College of Medicine and Health Science, Hawassa University, Hawassa, Ethiopia

* workuketema@hu.edu.et, workuketema@gmail.com

## Abstract

### Introduction

The global prevalence of Diabetes Mellitus (DM) has increased alarmingly over the last two decades. On top of this, the issues of non-adherence to the prescribed medicines further fuel the DM- related complications to become one of the top causes of mortality and morbidity. Despite the considerable efforts in addressing the poor adherence issues, there are still plenty of problems ahead of us yet to be addressed. The objective of this study was to determine the extent of non-adherence and its contributing factors among diabetic patients attending the medical Referral clinic of Dilla University Referral Hospital.

### Methods

The institutional-based descriptive cross-sectional study was carried out among patients with diabetes mellitus attending the medical referral clinic of Dilla University Referral Hospital.

A systematic random sampling method was used to recruit study participants, and tool was adopted to assess for adherence. A pretested semi-structured questionnaire was used to collect information on factors influencing non-adherence to the diabetic medications, and in-depth interview questionnaire was used for key informant interviews for the qualitative part. Data analysis was carried out using SPSS-20.

### Results

The overall prevalence of non-adherence to diabetic treatment regimen among the study participants was 34.0%. The study revealed that cost of transport to the hospital and taking alcohol were significantly associated with non-adherence to the diabetic treatment regimen

**Data Availability Statement:** All relevant data are within the manuscript and its Supporting information files.

**Funding:** This study was performed under the Ministry of Science and Higher Education and supervision of Dilla University College of Medicine and Health Sciences.

**Competing interests:** The authors have declared that no competing interests exist.

with the (AOR = 6.252(13.56, 28.822); $p < 0.000$) and (AOR = 13.12(8.06, 44.73); $p<0.002$) respectively.

## Conclusions

The study revealed that significant numbers of participants were non-adherent to the Diabetes Mellitus treatment regimens. Intensive counseling, and health education on the importance of good adherence and negative consequences of poor adherence need to be discussed with the patients before starting the medications, and amidst follow up.

## 1. Introduction

The leading endocrine disorder, Diabetes mellitus, is characterized by an abnormal metabolism and an inappropriately raised amount of serum glucose due to either absolute shortage of insulin or reduced tissue responsiveness to insulin [1].

The total number of people who will be affected by diabetes is expected to be 366 million cases by the year 2030. The rate of increment of this disease is immense in developing countries because of changes in lifestyle over the last few years. The region is being affected by the dual effects of non-infectious diseases like DM and infectious diseases, yet with the problems of accessibility to health care, and treatment [2,3].

Even though data regarding the prevalence of Diabetes Mellitus in active Tuberculosis is not sufficient in our country, literatures in other countries show the significant burden of Diabetes mellitus among active Tuberculosis with negative health outcomes. The problem is more worrisome in the developing countries where Tuberculosis is endemic, and the expected prevalence of Diabetes Mellitus is increasing alarmingly [4,5].

Diabetes Mellitus is the most important cause of both mortality and early infirmity; in the United States, it is the leading cause of blindness among working-age adults, end-stage renal disease, and non-traumatic limb amputations. It also increases the risk of cardiac, cerebral, and peripheral vascular disease two- to seven-fold and is a major contributor to neonatal morbidity and mortality and in the obstetric setting. Most of the overwhelming complications of diabetes can be prevented or delayed by the proper treatment of increased blood glucose levels and other modifiable risk factors. In treating diabetes, the timing of therapy is fundamental as the early recognition and treatment of the disease decides the clinical outcome [2,3,6].

In Africa, up to 80% of diabetic patients are underdiagnosed, and appearing to the health care facilities with complications is not uncommon. The chronic nature of the disease without symptoms will contribute for the late presentations. For some, poverty is the main reason for not to appear at Hospitals and take medications [7,8].

In Ethiopia, the national data on prevalence and incidence of diabetes are unsatisfactory despite an increment in patient attendance rates and medical admissions in the major hospitals. The World Health Organization (WHO) projected the number of diabetic cases in Ethiopia by 2030 to about 1.8 million [9].

Apart from the poor health seeking behavior of the society, non-adherence to the management play a pivotal role in the development of diabetes related complications. If the patient is not a good adherent, complications like cardiovascular, renal, neurologic [6,10], and extra cost to the health care system [11–13] will result. The cumulative global expenditure on the management of diabetes and its complications in 2013 was US$ 548 billion. In Africa, the cost per

person with diabetes was US$ 208.07 (IDF, 2014a). This emphasizes the serious need of attention in the prevention and management of diabetes, and also addressing issues of non-adherence [14–16].

In Ethiopia, the pervasive problems of economic instability, low educational background, and service unreachability to health care facilities might have played a significant role in the increased incidence of medication non-adherence. As far as our knowledge goes, the information on the level of non-adherence to diabetic treatment and the associated factors lacks in the study setting. We believe it was necessary to establish the factors that affect the adherence to the anti-diabetic treatment with the goal of improving the adherence and ultimately decreasing the incidences of diabetes complications.

## 2. Methodology

### 2.1 Study setting

The study setting was Dilla University Referral Hospital, Gedio zone, Southern Nation Nationality Peoples Regional State, Ethiopia. Dilla University Referral Hospital found in Dilla town that is located 365 km south of Addis Ababa, (the capital city of Ethiopia). The Hospital was established In 1928 G.C the Sudan Interior Mission (SIM) established a one block clinic in Dilla, Later in 1958 G.C, this clinic was upgraded and named "Leul Mekonnin Hospital" After 48 years of inception (in 1976 G.C) the missionaries handed the hospital over to the Ministry of Health (MOH). The hospital was upgraded to give services for 250, 000 population and inaugurated on January 30, 1985 G.C bearing the name "Dilla Rural Hospital'.' June 2009 G.C, it has been transferred to the hands of Dilla University in accordance with the agreement signed between Dilla University and Gedeo Zone Administration.

Currently, it is under the college of medicine and health science of Dilla University that gives educational service for new innovative medical education program students, Emergency Surgery & Obstetric, Health Officer, Midwifery, Psychiatry, and Anesthesia together with clinical service to the community. The hospital gives services to about 5 million people in Southern part of Southern Nation Nationalities and People and Southern parts of Oromia and Somali. Now the hospital has 555 different workers who are functioning on different services, of which 335 are administration workers. The other 220 workers are health professionals.

### 2.2. Study design and population

Institutional based cross-sectional study was conducted in order to assess non-adherence to diabetic treatment regimen and factors associated with non-adherence to treatment regimen among diabetic out patients at Dilla University Referral Hospital who came for follow up at Diabetic and chronic care clinic.

### 2.3. Inclusion & exclusion

**2.3.1 Inclusive criteria.** Those patient diagnosed as diabetes mellitus in chronic follow-up for greater than three months Patients greater than 18 years Avail themselves during data collection time.

**2.3.2. Exclusive criteria.** Very ill patients/admitted inpatient were excluded, Patient age less than 18 years and Those who are in diabetic coma or mentally incompetents.

Gestational diabetes

## 2.4. Sample size determination

The sample size was calculated using single population proportion formula as follows

$$n = \frac{Z_1^2 P(1-P)}{w^2} \quad n = \frac{(1.95)^2 0.5(1-0.5)}{(0.05)^2}$$

$$n = \frac{3.84 \times 0.5 \times 0.5}{0.025} = 384$$

Where:

**n** = desired sample size for population >10,000.

**Z** = standard normal duration usually set as 1.96 (which corresponds to 95% confidence level).

**P** = we use positive prevalence estimated. To maximize sample size. Negative prevalence = 1–0.5 = 0.5.

**W** = degree of accuracy desired (marginal error is 0.05).

As there is no previous study on topic under study in the study area, to estimate prevalence a figure of 0.5 used to get the possible minimum large sample size.

Since the total population is<10,000 that is 318; we use the Correction formula to determine final sample size.

$$nf = \frac{n}{1 + \frac{n}{N}} = nf = \frac{384}{1 + \frac{384}{318}} = 175$$

N = final sample size when a population is <10,000.

n = initial sample size when the population is >10,000.

nf = estimated study population.

Then 10% contingency was added on 175

$$175 \times 10\% = 17.5 = nf + contingency = 191$$

## 2.5. Study design, participants, and sampling procedure

The institutional -based cross-sectional study was conducted in order to assess non-adherence to diabetic treatment regimens and factors associated with non-adherence to treatment regimens among diabetic outpatients. All diabetic patients who attend the medical referral clinics for treatment and follow up at Dilla University Referral Hospital were included. The study population consisted of all diabetic patients who attend the chronic illness clinics for treatment and follow up of Dilla University Referral Hospital during the study period were included. The single population proportion formula was used to calculate the sample size.

For the quantitative part, the study participants were recruited as they came to the clinic using a systematic random sampling technique with a sampling interval of every third patient. The sampling was done until 191 study participants were recruited. The choice of the first study participant was identified through random sampling by picking either the 1st or the 2nd patient by way of tossing a coin. Participants who met the inclusion criteria and willing to participate in the study were included. For the qualitative part, purposive sampling was used to select nurses and doctors as key informants from among nurses and doctors working at the diabetic clinic.

The quantitative data was collected using a pretested semi-structured questionnaire which was used. It was serialized and administer to the study participants with the help of 3 trained

research assistants, a supervisor, and the principal investigator. The data included demographic characteristics, assessment of non-adherence, and reasons for non-adherence.

Adapted version of Morisky Medication adherence questionnaire was developed to evaluate non-adherence to diabetic treatment [17]. Based on the scores obtained 0 was considered high adherence, 1 or 2 as medium adherence, and >2 was low adherence. In this study, medium and high adherence were considered as adherent and low adherence as non-adherent for statistical purposes. The study participants were asked to recall whether they had missed any doses of any ant-diabetic medications on day to day basis over the last week. To increase accuracy on the number of pills prescribed, the study participants' hospital files and previous prescriptions were reviewed. Study participants were investigated for reasons for non-adherence to diabetic treatments. For physical exercise and dietary assessment, whether the patient is doing regular exercise and follow dietary advice given by his treating physician, the treating physicians were sources of information.

The questionnaires were checked for completeness and reliability of responses manually. For the quantitative part, data were coded, entered in SPSS for windows version 20.0. Appropriate descriptive and analytical (Chi-square, OR, bivariate, multivariate) test was used to determine the prevalence of non-adherence and statistically significant association between the dependent variable and independent variables whereas for the qualitative part thematic approach was used. Results were presented in texts, graphs, and tables.

### 2.6. Study variables

**2.6.1. Dependent variable.**   Non-adherence to Diabetes mellitus treatment.
**2.6.2. Independent variables.**   *Socio-demographic Variable.*

- Sex

- Age

- Religion

- Marital status

- Monthly income

- Occupation

- Social support

- Educational status

  **Health service factor- patient related factors**.

- Distance from the hospital

- Drug supply

- Staff motivation

- Diabetic education

  **Treatment factors**.

- Side effects

- Duration treatment

- Duration since diagnosis

- Pill burden/ Run out of pills

- Perception on the causes of diabetes

- Perceived benefits of treatment

- Perceived harm of treatment

- Other chronic illness

- Use of traditional medicine

- Monitoring blood sugar level

### 2.7. Data quality assurance

The quality of data was assured by properly designed questionnaire, proper training of the interviewers and supervisors of the data collection procedures, proper categorization and coding of the questionnaire. Every day, 10% of the computed questionnaires are viewed and checked for completeness and relevance by the supervisors and principal investigator and the necessary feedback was offered to data collectors in the next morning before the actual procedure.

### 2.8. Ethical considerations

IRB of Dilla University had approved the ethical clearance. Based on the objective of the study an official letter was sent to Dilla University Referral Hospital that was involved in the study from Dilla University, College of Health Science Research and Publication committee prior to the data collection period. Verbal informed consent was obtained after briefing the objective of the study. Those who were willing to be interviewed were signed in the space provided. The minor groups were not included in our inclusion criteria. Confidentiality was maintained and all respondents' questionnaire anonymously prepared S1 File.

## 3. Results

### 3.1. Socio-demographic and economic characteristics

A total of 191 diabetes mellitus patients were enrolled in this study, out of which 106(56%) were males and 85 (44%) were females. The mean age of the study participants was 55.4 (SD ± 12.85). Most 114(60.2%) of the study participants were aged between 40–59 years and a few 21(11.5%) were aged less than 40 years. About 63(33.5%) participants could only read and write and 55(29%) had attended grades 1–8 whereas 15.2%, 10.5% had no formal education and college/university respectively. More than 80% of the study participants were married and 13.2% were widowed with 4.2% divorced. The occupation section of the respondents indicates that 32.5%, 23%, and 23.6% of patients were housewives, government employees and private working respectively. About 3.1% and 3.7% were retired and student respectively. Approximately half of the study 49.2% participants earned less than five hundred Ethiopian birr per month.

More than half of the study participants, 122(63.9%), had social support. Among them, 52 (42.6%) by a family member, 21(17.2%) by local edir and ekub, 18(14.8%) by the government workers like health extension workers, 16(13.1%) by faith- based organization, and the remaining 15(12.2%) were supported by the non-government organization Table 1.

**Table 1. Socio-demographic and economic characteristics of T2DM patients who attend the medical referral clinics for treatment and follow up at DURH, 2017.**

| Variable | Frequency | Percent |
| --- | --- | --- |
| **SEX** | | |
| Male | 107 | 56.0 |
| Female | 84 | 44.0 |
| **Age (Years)** | | |
| <30 | 18 | 9.4 |
| 30–40 | 35 | 18.4 |
| 41–50 | 50 | 26.2 |
| 51–60 | 73 | 38.2 |
| >60 | 15 | 7.8 |
| **Religion** | | |
| Orthodox | 78 | 40.8 |
| Protestant | 68 | 35.6 |
| Muslim | 19 | 9.9 |
| Catholic | 23 | 12.0 |
| others (specify) | 3 | 1.6 |
| **Educational status** | | |
| No formal education | 29 | 15.2 |
| Read and write | 64 | 33.5 |
| Grade 1–8 | 57 | 29.8 |
| Grade 9–12 | 11 | 5.8 |
| TVET | 10 | 5.2 |
| College/University | 20 | 10.5 |
| **Marital status** | | |
| Never married | 3 | 1.6 |
| Married | 155 | 81.2 |
| Widowed | 25 | 13.1 |
| Divorced | 8 | 4.2 |
| **Occupation status** | | |
| Government employee | 45 | 23.6 |
| Private working | 44 | 23.0 |
| Student | 7 | 3.7 |
| Merchant | 27 | 14.1 |
| House wife | 62 | 32.5 |
| Retired | 6 | 3.1 |
| **Monthly income** | | |
| <500 | 92 | 49.2 |
| 501–1000 | 36 | 17.8 |
| 1001–2000 | 27 | 14.1 |
| >2000 | 36 | 18.9 |

## 3.2. Types of Diabetes Mellitus & perceived causes

About 173 (90.6%) of study participants were type II. More than half of reported patient perceptions on causes of diabetes were consuming sugary diets 56(29.3%), stress 33(17.3%), genetics 27(14.1%), eating fatty foods 27(14.1%), being overweight or obese 26(13.6%), inadequate physical activity 19(9.9%) and It is a punishment from God for past sins 3(1.6%). The

perception of diabetic patients on Causes of diabetes mellitus was evaluated by the key informants as appropriate perception.

> "Few patients relate it to consuming sugar diet especially the old generation but most of them don't know what caused their illness."

> (Key informant 2 –doctor).

> "Relatively a lot of our patients actually know and will tell you my ancestral related family members. So they know it is hereditary factor, genetic factor that contributes."

> (Key informant 4- doctor).

### 3.3. Magnitude of non-adherence to diabetic treatment regimen

The overall prevalence of non-adherence to diabetic treatment regimen among the study participants was 34.0%. About 73(38.2%) of the study participant do not follow either the dosing drugs or appointment according to the agreement with the treating physician. The major reasons participants ascribed to these were nothing should be swallowed during the fasting period, ant diabetic drugs are not necessary if sugary diets were not consumed, traditional medicine cure diabetes, prayers could cure diabetes and forgetfulness.

The majority, 166(86.9%), of the study participants do not conduct a regular exercise to standard. Most of the participant reported that lack of information (70.2%), shortage of time (17.8%), the difficulty of changing previous habits (6%), granting self-permission(4.2%), poor self-control(1.3%), and critical illness (5%) were the main reason non-sticking to the exercise. Regarding diet management 79.6% of the participant did not attend to their dietary intake according to the advice given by the managing physician due to lack of money to buy food (52.4%), lack of diet options (27.2%) economic reasons S1 Fig.

Adherence to diabetic treatment regimen was viewed by the key informants to be insufficient. Between 20% and 40% of the target population was said to be non-adherent.

> "It is difficult to estimate in percent but I can say in general the adherence to diabetic treatment regimen is not good."

> (Key informant 1-doctor).

> "Adherence is insufficient among patients attending our medical referral clinic. I would say 20% do not adhere and 80% adhere."

> (Key informant 2- doctor).

> "I can say adherence is not a hundred percent. Those who adhere are 70% and those who don't are 30%."

> (Key informant 3 –nurse).

### 3.4. Patient-related factors for non-adherence diabetic treatment regimen

The predominant patient- related reasons reported by participants for missing to take diabetic treatment regimen were, when away from home 154 (80.6%), when taking alcohol 148(77.5%),

**Table 2. Patient related factors for non-adherence to diabetic treatment regimen among T2DM patients who attend the medical referral clinics for treatment and follow up at DURH, 2017.**

| Variables | Frequency | Percent |
|---|---|---|
| Miss Rx when away from home | 154 | 80.6 |
| Miss Rx when take alcohol | 148 | 77.5 |
| Long lasting comorbidity | 117 | 61.3 |
| Miss Rx when symptom are controlled | 50 | 26.2 |
| Miss Rx when felt worse | 35 | 18.3 |
| Miss Rx when difficulty of remembering | 26 | 13.1 |
| Miss Rx when upset/depressed | 29 | 15.1 |
| Miss Rx when Busy | 24 | 12.6 |
| when did not understand | 23 | 12 |
| when feel better | 18 | 9.4 |
| when feel medication harm health | 14 | 7.3 |
| Miss when feel No benefit | 6 | 3.1 |
| Fear of Stigma | 22 | 11.5 |

and any long-lasting comorbidity 117(61.3%) whereas few participants 6(3.1%) reported that they do take treatment when feeling better Table 2.

Most of the key informants quoted being away from home, change of habits, stopping to take medicine when they feel better, perceived lack of efficacy of the prescribed medicine, and forgetfulness as patient- related factor for non-adherence.

"Most of chronic care patient particularly diabetic patients being away from home and change of habits mostly observed problems on follow up"

(Key informant 2- doctor).

"Some of the patients complain forgetfulness as the main reason for missing the ordered medication"

(Key informant 3-nurse).

"Some patient misses their drug when they feel better and when they think lack of efficacy of the prescribed medicine"

(Key informant 2- doctor).

### 3.5. Health care system and treatment-related factors

More than half 122 (63.7%) of the study participants resided at distant less than or equal to 10 km away from the facility and about 71(37.1%) live at distant more than 10km. About (49) 25.7% of the participant missed the appointment due to inability to afford the transportation cost. Almost half of the participants did not get their diabetic education from the treating physician and also about one-third of the participants 57(29.6%) had a strained relationships with health care providers. Near half of the study participants 102(53.4%) obtained their diabetic medication(s) from Dilla University Referral hospital pharmacy.

Approximately 152(80%) missed diabetic medicine due to they cannot afford and more than half 111 (58.1%) were on co-medication for long-term illnesses such as anti-hypertensive.

**Table 3. Distribution of study participants by duration since the diagnosis of diabetes mellitus among T2DM patients who attend the medical referral clinics for treatment and follow up at DURH, 2017.**

| Duration in year since Diagnosis | Frequency | Percent |
|---|---|---|
| < = 2 | 55 | 28.8 |
| 2–5 | 39 | 20.4 |
| 5–10 | 47 | 24.6 |
| >10 | 50 | 26.2 |

A relatively high proportion, 128(66.8%) of study participants had at least one diabetes complication. About 11(6.0%) reported that they sometimes used traditional medicine for managing diabetes and again 26(13.6%) had experienced side effects like hypoglycemia.

About half of the study participants, 97(50.8%), were more than five years since the diagnosis and almost all of them 96(50.3%) has started diabetic treatment at the time of diagnosis confirmed. The majority, 183(95.8%), of the participant gets tested their blood sugar level Table 3.

Most of the key informants regarded occasional stock out of diabetic medication including diagnostic tests as major health system- related factors contributing to non-adherence to the diabetic treatment regimen.

> " Some patients say the drugs and blood sugar tests are no available in the hospital occasional or expensive when it is available and others complain that they do not have money for transport and even for food."

(Key informant 2—doctor).

> "Few patients say their drugs got finished and were waiting for the next clinic, others say they had traveled upcountry while others think they are okay."

(Key informant 3 –nurse).

## 3.6. Factors associated with non-adherence to diabetic treatment regimen

None of the socio-demographic and socio-economic characteristics was significantly associated with non-adherence to diabetic treatment regimen at ($p>0.05$). And also none of the patient perceptions on causes of diabetes mellitus was significantly associated with non-adherence to the diabetic treatment regimen ($p> 0.05$).

Alcohol usage, being away from home, being upset, feeling no benefit, stigma, and did not understand about treatment were statistically significant by bivariate analysis for COR at 20.02 ($p < 0.000$), 26.02($p< 0.000$), 3.63($p<0.042$), 2.68 ($p<0.002$), 4.04 ($p<0.046$) respectively. The rest of patient-related factors were not statistically significant ($p > 0.05$). Again the cost of transport to hospital and Side effect of treatment were statistically significant COR at 6.252 ($p < 0.05$) and 2.64 ($p<0.042$) respectively.

Variables from bivariate analysis with $p \leq 0.25$ were fitted into the binary logistic regression model to identify factors independently associated with non-adherence to the diabetic treatment regimen. The variables included were being away from home, taking alcohol, feeling better, being upset, and cost of transport to the hospital. Using the stepwise forward likelihood ratio method, the variables cost of transport to the hospital and taking alcohol were identified as the predictors of non-adherence to the diabetic treatment regimen. Cost of transport to hospital was significantly associated with non-adherence to the diabetic treatment regimen

**Table 4. Association between variable understudy and non-adherence diabetic treatment regimen among T2DM patients who attend the medical referral clinics for treatment and follow up at DURH, 2017.**

| Variables | Adherence level | | OR (95% CI) | |
|---|---|---|---|---|
| | Utia Adherent | Non-adherent | Cr COR(95% CI) | A AOR(95% CI) |
| **Alcohol usage** | | | | **** |
| Yes | 62(49.2%) | 64(50.8%) | **20(5.96,67.15)** | **13.12 (8.06, 44.73)** |
| No | 3(4.4%) | 65(95.6%) | 1.01 1 | 1 |
| **Being away from home** | | | | |
| Yes | 63(47.7%) | 69(52.3%) | **26(6,09–111.02)** | |
| No | 2(3.3%) | 57(96.7%) | 1 | |
| **Being upset** | | | | |
| Yes | 18(60%) | 12(40%) | 1. 3.63 (1.62,8.14) | |
| No | 65(33.6%) | 128(66.6%) | **1** | |
| **No benefit** | | | | |
| Yes | 10(55.6%) | 8(44.4%) | **2.68 (1.03,7.16)** | |
| No | 55(31.8%) | 118(68.2%) | **1** | |
| **Fear of stigma** | | | | |
| Yes | 14(63.6%) | 8(36.4%) | **4.04(21.60, 10.19)** | |
| No | 51(30.2%) | 118(69.8%) | **11 1** | |
| **Did not understand about treatment** | | | | |
| Yes | 13(56.5%) | 10(43.5%) | **2.90(1.19, 7.04)** | |
| No | 52(32.1%) | 116(67.0%) | **1** | |
| **Lack money for transport** | | | | |
| Yes | 61(95.3%) | 3(4.7%) | **625.6.25(13.56,28.82)** | **62 6.25(13.5,28.82)** |
| No | 4(3.1%) | 123(96.9%) | **3.63 1** | 1 |
| **Side effect** | | | **2** | |
| Yes | 11(55%) | 9(45%) | **2.64(1.03,6.76)** | |
| No | 54(31.6%) | 117(68.4%) | **11 1** | |

(AOR = 6.252(13.56, 28.822). Taking alcohol was significantly associated with non-adherence to diabetic treatment regimen (AOR = 13.12(8.06, 44.73) Table 4.

## 4. Discussion

In this study, there was high a prevalence of non-adherence to physical exercise and dietary regimen that is 86.9% and 79.6% respectively in comparison with adherence to ant diabetic medication. This is almost comparable with the studies carried out in Zimbabwe, Saudi Arabia, Mexico, and Hungary showed that 26%–85% of the study subjects did not follow the physician's advice on exercise however, the instructions on diet were followed by 38%–76.8% of them [18–21]. The reasons for non-adherence to diet recommendations could be lack of information, economic reasons, being away from home, Difficulty of changing previous habits, granting self-permission, and poor self-control. With regards to exercise, it could be attributed to lack of motivation, change in their habits, lack of information, exercise as potentially exacerbating illness, lack of exercise partner, and specific locations away from home.

The study has shown that, none of the social demographic characteristics were significantly associated with non-adherence to the diabetic treatment regimens. The findings are consistent with results from other studies from Zimbabwe and México [18,19]. The findings were in differences with studies carried out in Uganda [19,22] in which female gender was significantly associated with non-adherence to the diabetic treatment regimen and in South Western

Nigeria [23] where gender and occupation were significantly associated with non-adherence to the diabetic treatment regimen. This might be a result of not adjusting for confounding factors for the Nigerian study. The findings also differed from those studies carried out in [21] Hawaii [24] and France [22,25] where adherence was strongly associated with age. Presumably, the differences in sample sizes, might attribute to the discrepancies.

The study revealed that, the financial problem in which approximately 92 (50%) of the participants with a monthly income of less than 500 Ethiopian birr was one of the main external challenges of adherence. This is close to a study done in Nigeria which was around 56.6%. In Ethiopia, the non-adherence is 37.1% which was due to financial difficulty [23,26].

The most commonly reported diabetic patient's perceptions on causes of diabetes were consuming sugary diets, stress, genetics, eating fatty foods, being overweight or obese, and inadequate physical activity in the current study. The findings were consistent with those of other studies [10].

The Support provided by the family played a beneficial role in enhancing adherence, in our study around 42.6% was found while a similar result of 45.7% was found in another study The general finding from different research articles showed that patients who had emotional support and help from family members, or healthcare providers were more likely to be adherent to the treatment. [13,15]

In this study, it is also found that having social support have positively affected the adherence to diabetic treatment regimens. This finding is indifferent to the studies referred here [18,22]. This could be due to differences in lifestyles among populations in various countries whereby some could be more interrelated and supportive than others.

In this study, none of the patient perceptions on causes of diabetes mellitus were significantly (P > 0.05) associated with non-adherence to the diabetic treatment regimens in the current study. The finding was in line with a study carried out in Zimbabwe [18] but differs from a study conducted in the United Kingdom [27], this could have been due to differences in their socio-economic status.

Duration of having diabetes and long-standing other illnesses were not significantly associated with non-adherence to the diabetic treatment regimen in the current study. This finding is similar to the study done at around Jimma [28]. Duration of treatment, taking of traditional medicine, other medications for other long term illness, and side effects of the drugs were not significantly associated with non-adherence to diabetic treatment agent. The long duration of diabetes treatment greater than ten years was found to be significantly associated with non-adherence to oral hypoglycemic medications in a study conducted in Zimbabwe [18] which did not agree with the current study.

The health care system-related factors specifically the availability of diabetic medication, distance from health facility, patient-health care provider relationship, diabetic education, and high medication cost were not significantly associated with non-adherence to the diabetic treatment regimen. This finding was almost similar to the study carried out in Jimma [28] in regard to the distance from the health facility but different from a study conducted in Zimbabwe. [18] This could be due to the difference in socioeconomic variation between Ethiopia and Zimbabwe for accessibility of health service [29].

In the current study, the cost of transport to the hospital and taking alcohol were identified as the predictors of non-adherence to the diabetic treatment regimens. Cost of transport to hospital was significantly associated with non-adherence to the diabetic treatment regimen (AOR = 8.51; 95% CI: 5.63–36.03; P < 0.000). Taking alcohol was significantly associated with non-adherence to the diabetic treatment regimen. (AOR = 4.12; 95% CI: 1.26–8.73; P < 0.002). Taking alcohol will affect the timing, and also contributes to forgetfulness [19,23,30,31].

## Conclusion

In this study, the overall prevalence of non-adherence to diabetic treatment regimen was substantial. Counseling the patients on the importance of adherence to the treatment is crucial.

## Supporting information

**S1 Fig. Showing the magnitude of adherence and non-adherence among the participants among T2DM patients who attend the medical referral clinics for treatment and follow up at DURH, 2017.**
(PDF)

**S1 File.**
(PDF)

## Acknowledgments

The authors acknowledge all the respondents who took part in the study.

## Author Contributions

**Conceptualization:** Bisrat Desalegn Boshe, Getachew Nenko Yimar, Aberash Eifa Dadhi, Worku Ketema Bededa.

**Data curation:** Bisrat Desalegn Boshe, Aberash Eifa Dadhi.

**Formal analysis:** Bisrat Desalegn Boshe, Aberash Eifa Dadhi, Worku Ketema Bededa.

**Funding acquisition:** Bisrat Desalegn Boshe.

**Investigation:** Bisrat Desalegn Boshe.

**Methodology:** Bisrat Desalegn Boshe, Getachew Nenko Yimar, Aberash Eifa Dadhi, Worku Ketema Bededa.

**Project administration:** Bisrat Desalegn Boshe.

**Resources:** Bisrat Desalegn Boshe.

**Software:** Bisrat Desalegn Boshe, Aberash Eifa Dadhi.

**Supervision:** Getachew Nenko Yimar.

**Validation:** Bisrat Desalegn Boshe, Getachew Nenko Yimar, Aberash Eifa Dadhi, Worku Ketema Bededa.

**Visualization:** Bisrat Desalegn Boshe.

**Writing – original draft:** Bisrat Desalegn Boshe, Getachew Nenko Yimar.

**Writing – review & editing:** Aberash Eifa Dadhi, Worku Ketema Bededa.

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
