## [Decision Letter · Decision Letter 0]

11 Jan 2021

PONE-D-20-36834

Assessing the Magnitude of Non-adherences to the treatment of DM and associated factors ; A hospital based Cross-sectional study

PLOS ONE

Dear Dr. Bededa,

Thank you for submitting your manuscript to PLOS ONE. After careful consideration, we feel that it has merit but does not fully meet PLOS ONE’s publication criteria as it currently stands. Therefore, we invite you to submit a revised version of the manuscript that addresses the points raised during the review process.

We look forward to receiving your revised manuscript.

Kind regards,

Claudia Marotta

Academic Editor

PLOS ONE

Additional Editor Comments:

dear authors follow reviewer suggestion to improve your paper

Journal Requirements:

2.) We note that your study uses the Morisky Medication Adherence Scale, which requires a license for use. Please include a short statement in your methods section to clarify whether you obtained a license to use the MMAS scale in your work, and whether the license was obtained prospectively or retrospectively.

3.) Please provide additional details regarding participant consent. In the ethics statement in the Methods and online submission information, please ensure that you have specified (1) whether consent was informed and (2) what type you obtained (for instance, written or verbal, and if verbal, how it was documented and witnessed). If your study included minors, state whether you obtained consent from parents or guardians. If the need for consent was waived by the ethics committee, please include this information.

4.) Please amend your list of authors on the manuscript to ensure that each author is linked to an affiliation. Authors’ affiliations should reflect the institution where the work was done (if authors moved subsequently, you can also list the new affiliation stating “current affiliation:….” as necessary).

5.) Please amend either the title on the online submission form (via Edit Submission) or the title in the manuscript so that they are identical.

6.) Please include captions for your Supporting Information files at the end of your manuscript, and update any in-text citations to match accordingly. Please see our Supporting Information guidelines for more information: http://journals.plos.org/plosone/s/supporting-information.

7.) Please include your tables as part of your main manuscript and remove the individual files. Please note that supplementary tables should be uploaded as separate "supporting information" files.

Reviewers' comments:

Reviewer's Responses to Questions

**Comments to the Author**

1. Is the manuscript technically sound, and do the data support the conclusions?

Reviewer #1: Yes

Reviewer #2: No

Reviewer #3: No

2. Has the statistical analysis been performed appropriately and rigorously? 

Reviewer #1: Yes

Reviewer #2: No

Reviewer #3: Yes

3. Have the authors made all data underlying the findings in their manuscript fully available?

Reviewer #1: Yes

Reviewer #2: Yes

Reviewer #3: No

4. Is the manuscript presented in an intelligible fashion and written in standard English?

Reviewer #1: Yes

Reviewer #2: No

Reviewer #3: Yes

5. Review Comments to the Author

Reviewer #1: I read with great interest the paper. I find it well wrote on important topic.

Only some suggestion

1. Introduction: diabetes is a risk factor for onset infectious diseases (es tuberculosis) and to worst outcome. Please add information about this relationship and how they influence in worst each other (see and cite doi: 10.4314/ahs.v17i3.20. PMID: 29085405; PMCID: PMC5656213; doi: 10.1186/s13104-018-3209-9. PMID: 29402317; PMCID: PMC5800087T doi: 10.1111/tmi.12704. Epub 2016 May 18. PMID: 27102229.)

2. Methods and result well wrote

3. Discussion add how is not only important diagnosis but care diabetes especially in low setting (see and citeDiabetes in active tuberculosis in low-income countries: to test or to take care? Lancet Glob Health. 2019 Jun;7(6):e707. doi: 10.1016/S2214-109X(19)30173-1. PMID: 31097272)

Reviewer #2: The authors have made interesting observations in this study. Nevertheless, the study is demerited for several reasons as detailed in my comments below.

1. The study is not powered by a statistically calculated sample size as it has applied random sampling method. The observations of the study cannot be extrapolated to the general population due to it’s cross-sectional design.

2. The study was conducted in 2017 and is communicated to a journal in 2020. This is a major concern as it limits the validity of the results for the recent times. The sample size is 191 subjects. To conduct this study in a hospital setting should not have been time consuming. I am concerned for reasons in delay in conducting and completing the study.

3. The salient findings of the study as mentioned in the abstract do not add any novel dimension for further research. I do not find any significance except for the fact that it is reported from a hospital in Ethiopia.

4. The manuscript is badly formatted. It appears that the authors did not proof check the manuscript and did not adhere to neat formatting norms on Word document. The manuscript appears un-appealing to the reviewers.

5. The authors have used erroneous English with too many grammatical errors and incomplete sentences. For eg “ In the section on introduction the line “In Africa, up to 80 % of diabetic patients are underdiagnosed and will complicate” I am sure that this manuscript was not proof checked.

6. The reference for Morisky Medication adherence questionnaire has not been cited.

7. Though the tables 1 and 2 have been cited, they do not appear in the main PDF of the manuscript. The tables are badly formatted and do not appeal to the reviewers.

8. There are no foot notes below tables which indicate the lapse in correct presentation of results.

9. Figures 1 and 2 are shoddy and needless.

10. References are badly formatted and do not comply to the journal’s specifications.

11. References 1 is unsuitable for a citation.

12. In short, this is a shoddy presentation of cross-sectional study with no robust validation. It would not worthwhile to make technical comments for a manuscript which does not comply to the basic requirements of a scientific manuscript. The authors need to seriously review the standard of writing manuscripts before communicating them to an international peer reviewed journal of significant impact factor. In view of the comments above, I deem it unsuitable for publication in Plos One in the present form.

Reviewer #3: The present manuscript encompasses a diabetes mellitus disease and non adherence . There are many articles on this subject in the literature.

The present work needs a thorough revision in material and methods, results and discussion. Diabetes Mellitus Type 2 disease is extremely common amongst adults and the non adherence. Was this a case control study? the inclusion/exclusion criteria need more details. The status of Diabetes disease need to be determined by laboratory exams. These information can be shown in a descriptive table.

It is also not clear if the number of subjects included in the study is supported by sample size calculations, as both diabetes and non adherence are common in the general population. Can this subject size support your conclusions?

In summary, the study design needs to be revisited to improve its results and its relevance.

6. PLOS authors have the option to publish the peer review history of their article (what does this mean?). If published, this will include your full peer review and any attached files.

Reviewer #1: No

Reviewer #2: No

Reviewer #3: No

---

## [Author Response · Author response to Decision Letter 0]

4 Feb 2021

Manuscript PONE-D-20-36834

Point- by- Point Rebuttal Letter

We really thank the academic editor, and all the three reviewers for their valuable comments on our manuscript.

Please kindly find below our response to each point raised by the academic editor and reviewers. We hope that we clearly addressed all of them, and that the manuscript will be now appropriate for publication. We bolded the comments, and highlighted the responses by green color.

Sincerely,

On behalf of all the four authors,

Worku Ketema Bededa

Journal Requirements:

1.) Please ensure that your manuscript meets PLOS ONE's style requirements, including those for file naming.

Thank you for your guide, and we have checked the templates and made the adjustments to meet the journal requirements.

2.) We note that your study uses the Morisky Medication Adherence Scale, which requires a license for use. Please include a short statement in your methods section to clarify whether you obtained a license to use the MMAS scale in your work, and whether the license was obtained prospectively or retrospectively.

You raised important point, and since it is difficult to secure the license for MMAS, we developed a tool based on the contents of MMAS from study done by Boon-How Chew N-HH, Mohd-Sidik Sherina (17)

3.) Please provide additional details regarding participant consent. In the ethics statement in the Methods and online submission information, please ensure that you have specified (1) whether consent was informed and (2) what type you obtained (for instance, written or verbal, and if verbal, how it was documented and witnessed). If your study included minors, state whether you obtained consent from parents or guardians. If the need for consent was waived by the ethics committee, please include this information.

We apologize for not including these all data in the original document, and really appreciate your patience. It is now well explained in the methodology part, under the subtitle 2.8.

4.) Please amend your list of authors on the manuscript to ensure that each author is linked to an affiliation. Authors’ affiliations should reflect the institution where the work was done (if authors moved subsequently, you can also list the new affiliation stating “current affiliation:….” as necessary).

Thank you for the comments, and corrected

5.) Please amend either the title on the online submission form (via Edit Submission) or the title in the manuscript so that they are identical.

Done

6.) Please include captions for your Supporting Information files at the end of your manuscript, and update any in-text citations to match accordingly.

Done

7.) Please include your tables as part of your main manuscript and remove the individual files. Please note that supplementary tables should be uploaded as separate "supporting information" files.

We again apologizes for this, and now amended.

We generally appreciate your cherished comments, and now feel the manuscript is appropriate for publication, as we tried our best in addressing all your productive comments point by point.

Reviewer #1: I read with great interest the paper. I find it well wrote on important topic. Only some suggestion 1. Introduction: diabetes is a risk factor for onset infectious diseases (es tuberculosis) and to worst outcome. Please add information about this relationship and how they influence in worst each other (see and cite doi: 10.4314/ahs.v17i3.20. PMID: 29085405; PMCID: PMC5656213; doi: 10.1186/s13104-018-3209-9. PMID: 29402317; PMCID: PMC5800087T doi: 10.1111/tmi.12704. Epub 2016 May 18. PMID: 27102229.).

Response; we are grateful to respond to your constructive comments, and we viewed the aforementioned papers, included some important points as deemed relevant.

For instances; Even though data regarding the prevalence of Diabetes Mellitus in active Tuberculosis is not sufficient in our country, literatures in other countries show the significant burden of Diabetes mellitus among active Tuberculosis with negative health outcomes. The problem is more worrisome in the developing countries where Tuberculosis is endemic, and the expected prevalence of Diabetes Mellitus is increasing alarmingly. [4,5]

2. Methods and result well wrote 3. Discussion add how is not only important diagnosis but care diabetes especially in low setting (see and cite. Diabetes in active tuberculosis in low-income countries: to test or to take care? Lancet Glob Health. 2019 Jun;7(6):e707. doi: 10.1016/S2214-109X(19)30173-1. PMID: 31097272)

o We appreciate for your review, on discussion part, yes we included the importance of cost effective strategies in curbing the burden of DM, especially in low income countries. We however did not discussed Diabetes in the active TB as it was not included in the result part. In fact it was mentioned in the introduction part as indicated above. We surely will consider it in our future research project.

Reviewer #2: The authors have made interesting observations in this study. Nevertheless, the study is demerited for several reasons as detailed in my comments below.

1. The study is not powered by a statistically calculated sample size as it has applied random sampling method. The observations of the study cannot be extrapolated to the general population due to it’s cross-sectional design.

Response; the reviewer has made interesting points, and as you all one of the limitations of cross-sectional study design is that the study cannot comfortably extrapolate to the general population, but still can gives you clue on the burden of the problem. We recommended that our finding be justified by a better sample size and study design. In general, we strongly believe that our findings will add something to the diabetes mellitus care with all the limitations.

2. The study was conducted in 2017 and is communicated to a journal in 2020. This is a major concern as it limits the validity of the results for the recent times. The sample size is 191 subjects. To conduct this study in a hospital setting should not have been time consuming. I am concerned for reasons in delay in conducting and completing the study.

Response; Sure, we are somehow late in communicating the journal. This is mainly because I, the correspondent author, was somehow busy of my study, Residency. We agreed that the findings in this paper will contributes something for the Diabetes care, particularly as there is no study done in the area on specific topic so far.

3. The salient findings of the study as mentioned in the abstract do not add any novel dimension for further research. I do not find any significance except for the fact that it is reported from a hospital in Ethiopia.

This is an important point, and we still consider our study deemed important as it is the first in its kind in our study area. We, as a clinicians are witnessing the problems of adherence with Diabetes treatment in particular, and chronic diseases in general. We therefore want to know what is really behind these all mess specifically in our area. We are at least come to know that alcohol consumption and distance from the health facilities are among the obstacles, hence interested to work on these area for the betterment of our patients.

4. The manuscript is badly formatted. It appears that the authors did not proof check the manuscript and did not adhere to neat formatting norms on Word document. The manuscript appears un-appealing to the reviewers.

We apologize for these inconveniences, and hope that you get the manuscript perfectly interesting now than before.

5. The authors have used erroneous English with too many grammatical errors and incomplete sentences. For eg “In the section on introduction the line “In Africa, up to 80 % of diabetic patients are underdiagnosed and will complicate” I am sure that this manuscript was not proof checked.

We thank the reviewer for his effort, and we amended it, after we sought help from our colleges English Department staffs. It is now edited as; “In Africa, up to 80 % of diabetic patients are underdiagnosed, and appearing to the health care facilities with complications is not uncommon”.

6. The reference for Morisky Medication adherence questionnaire has not been cited.

We appreciate for your detail evaluation of the paper, and now corrected and referred as reference number (17).

7. Though the tables 1 and 2 have been cited, they do not appear in the main PDF of the manuscript. The tables are badly formatted and do not appeal to the reviewers.

Done

8. There are no foot notes below tables which indicate the lapse in correct presentation of results.

We now put the amendment, and there is statement before each table elaborating about it.

9. Figures 1 and 2 are shoddy and needless.

Omitted now based on your recommendation.

10. References are badly formatted and do not comply to the journal’s specifications.

We understand and agree with this observation, and we thank the reviewer for pointing this out. It is now amended.

11. References 1 is unsuitable for a citation.

Corrected

12. In short, this is a shoddy presentation of cross-sectional study with no robust validation. It would not worthwhile to make technical comments for a manuscript which does not comply to the basic requirements of a scientific manuscript. The authors need to seriously review the standard of writing manuscripts before communicating them to an international peer reviewed journal of significant impact factor. In view of the comments above, I deem it unsuitable for publication in Plos One in the present form.

We really thank you for you’re the details and valuable comments on the manuscript, and we are optimistic that, now responses we replied above will suited you very well, and the manuscript is appropriate for publication.

Reviewer #3: The present manuscript encompasses a diabetes mellitus disease and non adherence . There are many articles on this subject in the literature. The present work needs a thorough revision in material and methods, results and discussion. Diabetes Mellitus Type 2 disease is extremely common amongst adults and the non adherence. Was this a case control study? the inclusion/exclusion criteria need more details. The status of Diabetes disease need to be determined by laboratory exams. These information can be shown in a descriptive table. It is also not clear if the number of subjects included in the study is supported by sample size calculations, as both diabetes and non adherence are common in the general population. Can this subject size support your conclusions? In summary, the study design needs to be revisited to improve its results and its relevance.

o Response; we thank the reviewer for his kind comments and useful insights. The methodology, results and discussion have been revised thoroughly. We included some important points which were not present in the original manuscript from the mother document.

o Regarding the study design, it was institutional based cross-sectional study, not a case control. The inclusion and exclusion criteria have been included in the methodology part.

o We understand your concern, and the study units were people with already diagnosed T2DM who have been on the treatment for more than three months.

o The detail of study design and sample size calculation was included under the methodology part, we would have included it in the sample size calculation. For further information, it has been described as follows;

o Sample size determination: The sample size was calculated using single population proportion formula as follows

n=Z12P(1−P)w2 n=(1.95)20.5(1−0.5)(0.05)2

n=3.84 x 0.5 x 0.50.025 n= 384

Where:

n=desired sample size for population >10,000

Z=standard normal duration usually set as 1.96 (which corresponds to 95% confidence level)

P=we use positive prevalence estimated. To maximize sample size. Negative prevalence =1-0.5=0.5

W=degree of accuracy desired (marginal error is 0.05)

As there is no previous study on topic under study in the study area, to estimate prevalence a figure of 0.5 used to get the possible minimum large sample size.

Since the total population is<10,000 that is 318; we use the Correction formula to determine final sample size.

=n1+nN = ��=3841+384318 = 175

N=final sample size when a population is <10,000

n=initial sample size when the population is >10,000

nf=estimated study population

Then 10% contingency was added on 175

175×10%=17.5 = nf + contingency = 191

We generally appreciate your valuable comments, and now feel the manuscript is appropriate for publication, as we tried our best in addressing all your constructive comments point by point.

---

## [Decision Letter · Decision Letter 1]

17 Feb 2021

The magnitude of non-adherence and contributing factors among adult outpatient with Diabetes mellitus in Dilla University Referral Hospital, Gedio, Ethiopia

PONE-D-20-36834R1

Dear Dr. Worku,

We’re pleased to inform you that your manuscript has been judged scientifically suitable for publication and will be formally accepted for publication once it meets all outstanding technical requirements.

Kind regards,

Claudia Marotta

Academic Editor

PLOS ONE

Additional Editor Comments (optional):

dear Authors congratulations

Reviewers' comments:

Reviewer's Responses to Questions

**Comments to the Author**

1. If the authors have adequately addressed your comments raised in a previous round of review and you feel that this manuscript is now acceptable for publication, you may indicate that here to bypass the “Comments to the Author” section, enter your conflict of interest statement in the “Confidential to Editor” section, and submit your "Accept" recommendation.

Reviewer #1: All comments have been addressed

Reviewer #3: All comments have been addressed

2. Is the manuscript technically sound, and do the data support the conclusions?

Reviewer #1: Yes

Reviewer #3: Yes

3. Has the statistical analysis been performed appropriately and rigorously? 

Reviewer #1: Yes

Reviewer #3: Yes

4. Have the authors made all data underlying the findings in their manuscript fully available?

Reviewer #1: Yes

Reviewer #3: Yes

5. Is the manuscript presented in an intelligible fashion and written in standard English?

Reviewer #1: Yes

Reviewer #3: Yes

6. Review Comments to the Author

Reviewer #1: Authors worte an interesting paper. The role of non comunicable diseases in low setting is crucial also in global halth approuch wiev

I suggest to accept this new version of paper

Reviewer #3: (No Response)

7. PLOS authors have the option to publish the peer review history of their article (what does this mean?). If published, this will include your full peer review and any attached files.

Reviewer #1: **Yes: **Francesco Di Gennaro

Reviewer #3: No

---

## [Editor Report · Acceptance letter]

22 Feb 2021

PONE-D-20-36834R1 

The magnitude of non-adherence and contributing factors among adult outpatient with Diabetes mellitus in Dilla University Referral Hospital, Gedio, Ethiopia 

Dear Dr. Bededa:

I'm pleased to inform you that your manuscript has been deemed suitable for publication in PLOS ONE. Congratulations! Your manuscript is now with our production department. 

Kind regards, 

on behalf of

Dr. Claudia Marotta 

Academic Editor

PLOS ONE